# Gene Expression Profiling of Fibroepithelial Lesions of the Breast

**DOI:** 10.3390/ijms24109041

**Published:** 2023-05-20

**Authors:** Xiaomo Li, Eric Vail, Horacio Maluf, Manita Chaum, Matthew Leong, Joseph Lownik, Mingtian Che, Armando Giuliano, Duoyao Cao, Farnaz Dadmanesh

**Affiliations:** 1Department of Pathology and Laboratory Medicine, Cedars-Sinai Medical Center, Los Angeles, CA 90048, USA; 2Saul and Joyce Brandman Breast Center, Samuel Oschin Comprehensive Cancer Institute, Cedars-Sinai Medical Center, Los Angeles, CA 90048, USA; 3Department of Biomedical Science, Cedars-Sinai Medical Center, Los Angeles, CA 90048, USA

**Keywords:** cancer, breast, fibroepithelial lesions, fibroadenoma, phyllodes tumor, molecular, gene expression, risk stratification, molecular genetics

## Abstract

Fibroepithelial lesions of the breast (FELs) are a heterogeneous group of neoplasms exhibiting a histologic spectrum ranging from fibroadenomas (FAs) to malignant phyllodes tumors (PTs). Despite published histologic criteria for their classification, it is common for such lesions to exhibit overlapping features, leading to subjective interpretation and interobserver disagreements in histologic diagnosis. Therefore, there is a need for a more objective diagnostic modality to aid in the accurate classification of these lesions and to guide appropriate clinical management. In this study, the expression of 750 tumor-related genes was measured in a cohort of 34 FELs (5 FAs, 9 cellular FAs, 9 benign PTs, 7 borderline PTs, and 4 malignant PTs). Differentially expressed gene analysis, gene set analysis, pathway analysis, and cell type analysis were performed. Genes involved in matrix remodeling and metastasis (e.g., *MMP9*, *SPP1*, *COL11A1*), angiogenesis (*VEGFA*, *ITGAV*, *NFIL3*, *FDFR1*, *CCND2*), hypoxia (*ENO1*, *HK1*, *CYBB*, *HK2*), metabolic stress (e.g., *UBE2C*, *CDKN2A*, *FBP1*), cell proliferation (e.g., *CENPF*, *CCNB1*), and the PI3K-Akt pathway (e.g., *ITGB3*, *NRAS*) were highly expressed in malignant PTs and less expressed in borderline PTs, benign PTs, cellular FAs, and FAs. The overall gene expression profiles of benign PTs, cellular FAs, and FAs were very similar. Although a slight difference was observed between borderline and benign PTs, a higher degree of difference was observed between borderline and malignant PTs. Additionally, the macrophage cell abundance scores and CCL5 were significantly higher in malignant PTs compared with all other groups. Our results suggest that the gene-expression-profiling-based approach could lead to further stratification of FELs and may provide clinically useful biological and pathophysiological information to improve the existing histologic diagnostic algorithm.

## 1. Introduction

Breast fibroepithelial lesions (FELs) are a heterogeneous group of biphasic neoplasms composed of both stromal and epithelial elements that range from indolent fibroadenomas (FAs) to malignant phyllodes tumors (PTs). They account for about 0.5–1.0% of all breast tumors [1]. Fibroepithelial lesions with cellular stroma are inconsistently classified as cellular FA or PT. Studies have shown high interobserver variability in the distinction between cellular fibroadenoma and benign phyllodes tumors, even among expert breast pathologists [2]. Moreover, despite World Health Organization published histologic criteria for distinguishing benign, borderline, and malignant phyllodes tumors [1], the overlapping morphology remains problematic [2,3]. From a management perspective, fibroadenomas can be followed and are usually treated with simple resection, whereas phyllodes tumors are treated with mastectomy or wide excision. The clinical behavior of FELs is highly variable; while malignant PTs have higher rates of recurrence and higher risk of distance metastasis, there are some reports of metastasizing benign and borderline PTs. Thus, within the current histological categorization, the malignant potential of the various FELs subtypes is difficult to predict based on histologic features alone [4,5,6,7].

In recent years, a wide range of studies have been performed on PTs and FAs with respect to their biomarker expression, genetic alterations, and molecular mechanisms, as potential tools for more reproducibly classifying PTs and FAs [8,9,10]. Gene expression profiling is an additional approach that may help predict the biological behavior of FELs and provide insights for their clinical management, though studies using this technique remain limited. Moreover, since malignant PTs have the potential to metastasize [11] and no effective targeted therapy has yet been reported, the finding of targetable genes/pathways for malignant PTs may also be of clinical use.

A variety of biomarkers, such as Ki-67 and p53 protein expression, have been proposed to help with subclassification of phyllodes tumors [12,13], and the expression of c-kit (*CD117*), phospho-histone3, *MDM2*, *ERG*, *CD31*, and *CD34* have been reported to be predictive of malignant behavior of PTs [12,14,15]. However, these ancillary studies have not been shown to be independent predictive and prognostic biomarkers [16]. In recent years, genomic sequencing has identified common mutations in fibroadenomas and phyllodes tumors such as *MED12*, *RARA*, and *MDM12* genes suggesting the same cellular origin [17,18,19]. However, the molecular alterations driving the pathogenesis of FELs are still poorly defined [5,20]. Moreover, detection of differentially expressed genes within the different subgroups of fibroepithelial neoplasms (FAs, benign, borderline, and malignant PTs) is limited. Thus, currently proposed molecular alterations and immunohistochemistry are of limited use in routine practice. Further studies to characterize the gene expression profiles of FELs may more reliably classify and stratify risk for this group of lesions.

To this end, we performed gene expression profiling across the different categories of breast fibroepithelial lesions, analyzing the expression of 750 tumor-related genes. Our aim was to characterize the gene expression profiles of the different subcategories of breast fibroepithelial lesions to determine whether gene expression profiling might aid in the distinction between these subcategories and provide information relevant to disease risk stratification in this category of breast lesions with overlapping histologic features. 

## 2. Results

### 2.1. Gene Expression Profiling among Fibroadenomas, Cellular Fibroadenomas and Benign Phyllodes Tumors

Fibroadenomas (FAs), cellular FAs, and benign PTs shared overlapping gene expression profiles, with very few differentially expressed (DE) genes identified. Volcano plots demonstrating the distribution of the fold changes in gene expression levels and *p*-values are shown in Figure 1. A few genes (*FSTL3*, *DUSP2 CDKN2A*) were slightly upregulated in cellular FAs in comparison to FAs. Only three genes (*SERPINA1*, *COMP*, *TWIST1*) were slightly differentially expressed in benign PTs when compared with cellular FAs and FAs.

### 2.2. Gene Expression Profiling Differentiates Malignant Phyllodes Tumors from Borderline Phyllodes Tumors, Benign Phyllodes Tumors and Fibroadenomas 

To visualize the difference in expression levels of the 750 cancer-related genes across the spectrum of FELs, we performed a hierarchical clustering of differentially expressed (DE) genes among all groups. Malignant PTs clustered distantly apart from all the other FELs (Figure 2A). A total of 131 DE genes (63 upregulated and 68 downregulated) were identified in malignant PTs, compared to the other FELs. Volcano plots showing pair-wise comparisons of differentially expressed genes between malignant PTs and each of the remaining FELs are shown in Figure 2B. While there were some differences in the gene expression profile between borderline and benign PTs (10 upregulated and 10 downregulated genes), more distinctive differential gene expression was observed between borderline and malignant PTs (30 upregulated and 26 downregulated genes).

### 2.3. Distinct Biological Processes Involved in Malignant PTs

To identify the biological processes that are altered in malignant PTs, gene set analysis was performed and pathway scores were defined (Figure 3). Expression of genes such as *ENO1*, *HK1*, *CYBB*, *HK2*, *VEGFA*, and *ERBB2*, which are hypoxia-related genes, was significantly altered in malignant PTs. Also, genes involved in various cancer-associated intracellular signaling cascades, including interferon signaling (*EIF2AK2*, *ICAM1*, *STAT2*, *IRF3*), NF-kappa B signaling (*NFKB1*, *IKBKG*, *IKBKB*), PI3K-Akt (*PIK3R1*, *PTEN*, *PRLR*) and TGF-beta signaling (*TGFB1*, *ID4*) were also differentially expressed in malignant PTs. Genes regulating varying biological processes, such as apoptosis (*CDH1*, *BIRC5*, *PSMB5*, *TNFSF10*), epigenetic regulation (*HDAC11*, *BNIP3*, *MAP2K12*), matrix remodeling, and metastasis (*MMP9*, *TGFB1*, *LOXL2*, *CDH1*, *PLOD2 COL11A1*, *COL5A*, *MMP7*), also showed distinctive expression patterns.

### 2.4. Macrophage Abundance Score Is Increased in Malignant and Borderline PTs 

Cell type profiling analysis measures genes that are widely accepted to be characteristic of various cell populations (dendritic cells, macrophages, neutrophils, T cells, NK cells, etc.). In the 34 FEL cases analyzed, macrophage abundance scores measuring macrophage-specific surface marker genes (CD68, CD84) were highest in malignant PTs, followed by borderline PTs (Figure 4A). CD163 immunohistochemistry also highlighted increased macrophage infiltrations in malignant PTs (Figure 4B).

## 3. Discussion

In this study, we measured and analyzed the expression of 750 tumor-related genes in 34 breast fibroepithelial lesions as a potential aid in their classification and to explore their underlying biology. Although fibroadenomas and benign phyllodes tumors are recognized as distinct entities, their common overlapping histologic features (particularly between cellular FA and benign PT) make the distinction challenging in daily practice. Several comprehensive genetic and proteomic studies of FELs suggest that FAs and benign PTs share similar gene and protein expression profiles [8,9,17,18]. Our study revealed that fibroadenomas (FAs), cellular fibroadenomas, and benign phyllodes tumors (PTs) manifest similar gene expression profiles (Appendix A). Genes that are involved in different cancer-related pathways, such as *MAPK*, epigenetic regulation, matrix remodeling and metastasis, angiogenesis, TGF-beta signaling, hedgehog signaling, and PI3K-Akt, showed no difference between these three groups when gene set analysis was performed. Similar patterns were also recognized in these three groups of FELs using cell type analysis. While the current treatment protocol for asymptomatic fibroadenomas is conservative management (such as annual breast examination and ultrasound as clinically indicated [21]), benign phyllodes tumors are treated with surgical intervention in many medical centers [22,23]. The overlapping histologic and gene expression profiles of FAs, cellular FAs, and benign phyllodes tumors could imply a similarly indolent clinical behavior and may suggest that a conservative management strategy is appropriate for each of these entities.

A recent study revealed that borderline phyllodes tumors could be separated into either benign or malignant PTs using molecular assays [24]. Interestingly, in our study, the hierarchical clustering heatmap shows several borderline phyllodes cases clustered with the malignant PTs group, and some clustered closely with the benign phyllodes tumor group (Appendix A). This suggests that a subset of borderline PTs exhibits a gene expression profile that overlaps with that of malignant PTs. Various gene alterations were found in both malignant PTs and borderline PTs (Appendix A), which could imply that some malignant PTs derive from a subset of borderline PTs through clonal evolution and/or acquisition of somatic mutations leading to malignant transformation. Prospective studies utilizing gene expression analysis with a larger sample size could aid in optimizing the diagnostic and prognostic stratification of borderline PTs. Additional studies are also required to determine whether the remaining subset of borderline PTs that lack a gene expression profile comparable to that of malignant PTs may also progress to malignancy via de novo pathways.

Extracellular-matrix (ECM)-related gene alterations are known to promote metastases in various cancers [25,26,27]. The ECM molecules are major components in the tumor microenvironment integral to cellular adhesion and harboring growth factors [28]. Fibroblasts secrete collagen, which is the main structural protein of the ECM responsible for intercellular adhesion, differentiation, and integrity [29]. Previous studies report that ECM gene dysregulation in genes such as, but not limited to, *COL1A1*, *FN1*, *TIMP1*, and *MMP9*, plays an important role in tumorigenesis in various malignancies and is associated with poor prognosis [27,28,30,31]. In this study, we also found higher expressions of collagen type I alpha 1 chains (*COL1A1*) as well as matrix metallopeptidases 7 and 9 (*MMP7*, *MMP9*) genes in the borderline and malignant PTs groups when compared with benign PTs. Further studies are required to construct and validate a comprehensive molecular panel composed of ECM-related genes that might aid with proper risk stratification of borderline PTs.

Our gene expression profiling results identified many important cancer-related biological processes, such as angiogenesis, matrix remodeling & metastasis, PI3K-Akt pathways, TGF-beta signaling, MAPK pathways, and hypoxia, that distinguish malignant PTs from other FELs. This is consistent with our current understanding of the pathogenesis and biologic behavior of malignant PTs, in which stromal overgrowth, increased angiogenesis, and extracellular matrix factors are profoundly involved in tumor progression [9]. Pathway scores and gene set analysis revealed that angiogenesis-related genes, including *VEGFA*, *ITGAV*, *NFIL3*, *FDFR1*, and *CCND2*, are highly upregulated in malignant PTs. Recent studies have demonstrated that VEGFRs are expressed in several cancer cell types and may dictate tumor invasion [32]. This finding could potentially predict a response to the targeted therapy (angiogenesis inhibitor). Overexpression of *EGFR* and *PDGFA* is also present in malignant phyllodes tumors, which suggests the potential use of dual VEGFR/EGFR inhibitors (Vandetanib), a combination of anti-VEGFR and anti-EGFR and multikinase inhibitors (Panzopnib).

The growth and survival of malignant cells are often driven by onstituteve activation in the mitogen-activated protein kinase (MAPK) and phosphor-inositide 3-kinase (PI3K/AKT) signaling pathways [33,34,35]. VEGFR-2 mediated activation of the PI3K/Alk cascade is also important for tumor survival. It is well established that no pathways exist independently; crosstalk between the pathways promotes tumor cell proliferation, survival, and invasion. Gene set analysis (GSA) revealed that 28 genes (*PIK3R1*, *PRLR*, *EIF4EBP1*, *NRAS*, *HRAS*, *MAP3K12*, *KIT*) involved in the PI3K/AKT and MAPK pathways were differentially expressed in malignant PT groups. Moreover, genes in the NK-kappa B (*IKBKB*, *KIBKG*, *NFKB1*) and NOTCH (*NOTCH1*, *HDAC11*, *APH1B*) signaling pathways are also differentially expressed. *PIK3CA*/*KRAS* and *HRAS* alterations have already been reported in borderline and malignant phyllodes tumors [36]. In contrast to the previous study, we identified higher frequency and more gene alterations in comparison with the previous data, which implies a potential therapeutic benefit from IKK inhibitors and mTOR inhibitors.

An increased macrophage abundance score in malignant phyllodes tumors was found in this cohort by cell type analysis. The score is calculated by measuring the expression level of four macrophage-related genes: CD163, CD68, CD84, and MS4A4A. We found CD68, CD84, and CD163 to be upregulated in malignant PTs. Many studies have found that breast cancer with high tumor-associated macrophage (TAM) infiltration was significantly correlated with aggressive biological behavior and could serve as an independent predictor of overall survival and recurrence-free survival [37,38]. Previous studies also proposed that TAMs could stimulate myofibroblast differentiation and promote proliferation as well as invasion of phyllodes tumors by the CCL18-driven NF-kB/PTEN/AKT axis [39]. The interactions of the tumor microenvironment and tumor cells have been shown to drive the progression of various cancers, and TAMs are the most abundant inflammatory cell type within the microenvironment of malignant PTs [40]. Additional TAM-associated markers, including CCL18, CCL5, MMP-9, and SPP-1, were also identified in our study by differential expression analysis [40]. Consistent with prior signaling pathway alteration results [40,41,42], our data suggest that TAMs play a crucial role in the tumorigenesis of malignant phyllodes tumors. However, identifying the specific pathways of TAMs interacting with myofibroblasts and other inflammatory cells leading to malignant progression would require further investigation.

Our analysis was limited by a relatively small cohort size. Future studies with a larger sample size are needed to validate our findings. In addition, correlating the molecular result with long term clinical follow-up data might provide relevant information to guide clinical management.

In conclusion, this study provided a comprehensive profile of gene alterations across various subtypes of breast fibroepithelial lesions and might bring new insights in their classification and improve understanding of their pathogenesis. Our results raise the possibility that fibroadenomas, cellular fibroadenomas, and benign phyllodes tumors might be merged in the same subgroup for the purpose of clinical management. Further studies would be of particular interest to predict the biological behavior of borderline phyllodes tumors.

## 4. Materials and Methods

### 4.1. Patient Sample Selection

This is a retrospective study. A total of 34 fibroepithelial lesions identified between 2017 to 2021 were selected from the pathology database at Cedars-Sinai Medical Center, including fibroadenomas (n = 5), cellular fibroadenomas (n = 9), benign phyllodes tumors (n = 9), borderline phyllodes tumors (n = 7) and malignant phyllodes tumors (n = 4). All H&E slides were blindly reviewed by two in-house senior breast pathologists who confirmed the diagnoses according to 2019 World Health Organization (WHO) criteria [1]. Clinical reports and follow-up data were available for all patients.

### 4.2. RNA Extraction and Gene Expression Analysis

All hematoxylin and eosin (H&E) stained slides were reviewed by two senior pathologists to select the most representative area for microdissection. Three 10 µm FFPE (formalin-fixed, paraffin-embedded) slides were cut for each case. Normal breast tissue was avoided during microdissection to eliminate contamination and signal dilution. Total RNA was extracted using the Rneasy FFPE kit (Qiagen, Hilden, Germany) and quantified with a Nanodrop Spectrophotometer (Thermo Fisher Scientific, Waltham, MA, USA), and 100 ng of total RNA was used to measure the expression of 750 tumor-related genes and 20 housekeeping/reference genes using the nCounter platform (Nanostring Technologies, Seattle, WA, USA) according to the manufacturer’s protocol. Expression counts were then normalized using the nSolver 4.0 software and custom scripts in R 3.3.2. False Discovery Rate [FDR] was used to identify differential gene expression across sample groups. Genes with at least 2-fold or higher expression were considered differentially expressed. Comparison of differential expression, gene set analysis, and cell type analysis was also performed using nSolver 4.0 software. Raw gene expression data has been deposited in Gene Expression Omnibus (GSE55224).

### 4.3. Immunohistochemistry

Immunostaining for CD163 was performed on selected sections from paraffin-embedded tissue blocks of each case and was produced in accordance with optimal staining standards. The slides were prepared at the immunohistochemistry lab at Cedars-Sinai Medical Center: 4-μm thick sections were incubated with monoclonal mouse antibody to CD163 (Leica clone 10D6, Leica Biosystems, Wetzlar, Germany). Staining was performed on the Roche Ventana Benchmark Ultra (Tucson, AZ, USA) automated slide stainer using an onboard heat-induced epitope retrieval method in a high pH buffer. Staining for CD163 was visualized using the Ventana ultraView DAB Detection Kit. All slides were subsequently counterstained with Mayer’s hematoxylin.

### 4.4. Statistical Analysis

Statistical differences for numerical values were calculated using the ANOVA test. The difference was considered statistically significant at a *p*-value of less than 0.05 (*p* < 0.05). All statistical analyses were performed using GraphPad Prism v9. (GraphPad Software, Inc, San Diego, CA, USA). Heatmaps were generated in R using the ComplexHeatmap package. Differentially expressed genes between samples were determined using Limma with a *p*-value of 0.05 using a Bonferonni correction. Clustering was conducted using genes that were differentially expressed among different subgroups using the standard clustering methods in the ComplexHeatmap package in R. All statistical tests were two-sided, and the statistical significance level was set to less than 0.05.

## Figures and Tables

**Figure 1 ijms-24-09041-f001:**
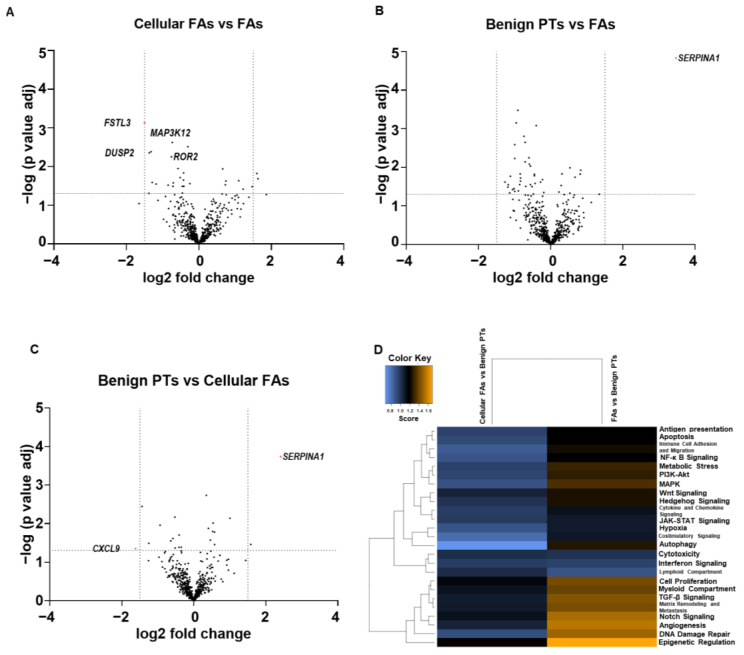
Gene expression profiles of fibroadenomas, cellular fibroadenomas and benign phyllodes tumors are similar. Volcano plots for pairwise comparison of (**A**) cellular FAs vs. FAs; (**B**) benign PTs vs. FAs; (**C**) benign PTs vs. cellular FAs. (**D**) Heatmap of global significance score shows no extensive upregulation or downregulation of the gene sets. (DE genes defined as >2-fold changes, adjusted *p*-value < 0.05).

**Figure 2 ijms-24-09041-f002:**
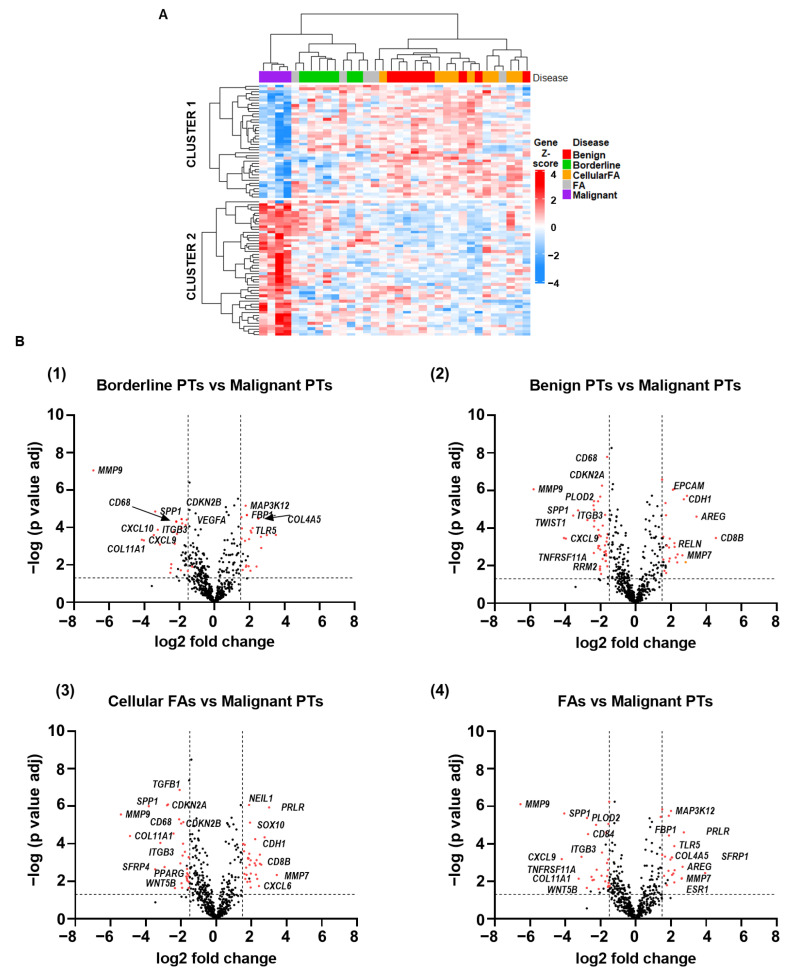
Malignant phyllodes tumors display a distinctive gene expression profile compared to all other breast fibroepithelial lesions. (**A**) Heatmap showing the hierarchical clustering of differentially expressed genes in patients with FELs (n = 34) based on NanoString PanCancer 360 panel. (**B**) Volcano plots show that large numbers of genes are differentially expressed in (B1) borderline PTs vs. malignant PTs; (B2) benign PTs vs. malignant PTs; (B3) cellular FAs vs. malignant PTs; (B4) FAs vs. malignant PTs.

**Figure 3 ijms-24-09041-f003:**
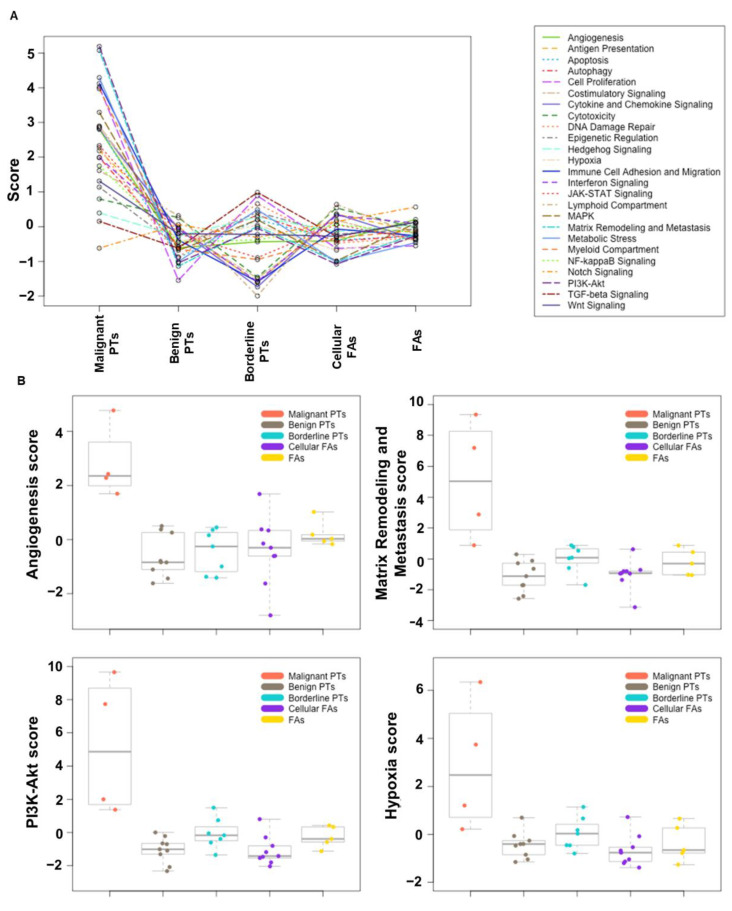
(**A**) Pathway analysis showed that gene sets in various biological processes are significantly altered in malignant PTs. A. Overview of all pathway scores vs. different groups of FELs. Lines show each pathway’s average score across values of different FELs groups. (**B**) Box and whisker plots for expression score of four cancer-related biological processes/pathways across five groups of FELs.

**Figure 4 ijms-24-09041-f004:**
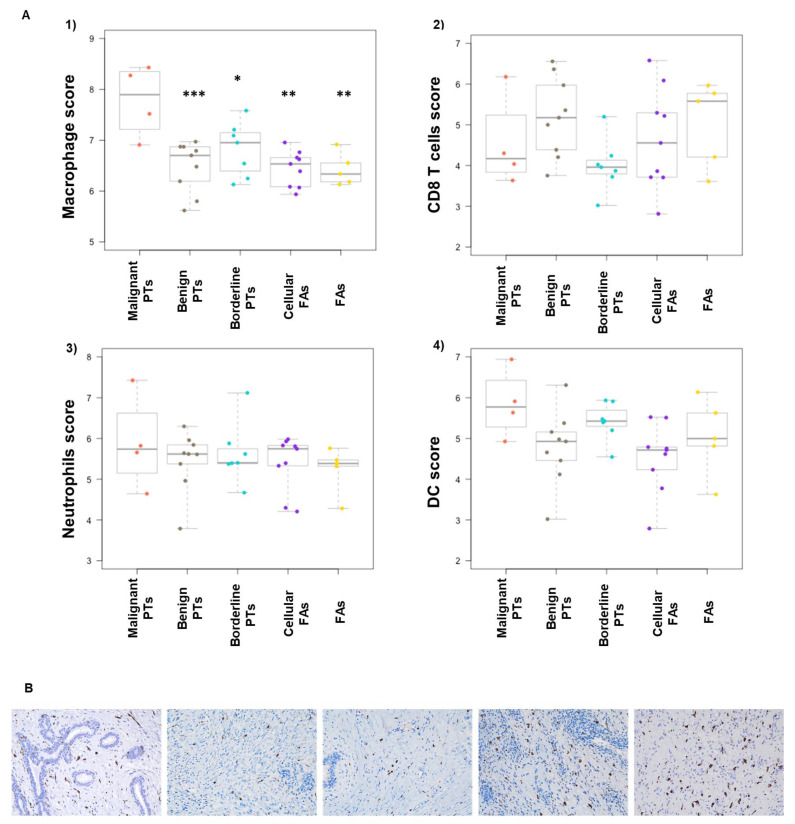
(**A**). Cell type analysis of five groups of FELs: (1) macrophage abundance score; (2) CD8 T cells abundance score; (3) neutrophils abundance score; (4) dendritic cell (DC) abundance score. * *p* < 0.05, ** *p* < 0.01, *** *p* < 0.001. (**B**). CD163 immunohistochemistry highlights macrophages in fibroadenoma, cellular fibroadenoma, benign phyllodes, borderline phyllodes and malignant phyllodes tumors (CD163, 20×).

## Data Availability

The datasets used and/or analyzed during the current study are available from the corresponding author upon reasonable request.

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
