# Peer review of "Gene Expression Profiling of Fibroepithelial Lesions of the Breast"

_ijms, 2023, doi:10.3390/ijms24109041_

Round 1

Reviewer 1 Report

1. In Figures 1 and 2, the text size in the volcano plots is too small and this is also true for the heatmap, making the text unreadable. 

2. Figure 4 is entirely missing. 

3. The authors specifically performed measurements of RNA levels for only 750 genes. The authors should provide a rationale for why they chose this set and the relevant citations for this set of genes. Further, why did the authors not choose to perform unbiased RNAseq protocol instead of selective measurement for 750 genes instead of the approx. 8-10k genes expressed. Since, most conclusions in this paper are based on just this small subset of genes.

No comment.

Author Response

1. In Figures 1 and 2, the text size in the volcano plots is too small and this is also true for the heatmap, making the text unreadable. 

Figures 1 and 2 have been updated to have increased readability

2. Figure 4 is entirely missing. 

Figure 4 has been added

3. The authors specifically performed measurements of RNA levels for only 750 genes. The authors should provide a rationale for why they chose this set and the relevant citations for this set of genes. Further, why did the authors not choose to perform unbiased RNAseq protocol instead of selective measurement for 750 genes instead of the approx. 8-10k genes expressed. Since, most conclusions in this paper are based on just this small subset of genes.

We chose to use the panel which was focused on cancer-related genes instead of the entire panel.

Reviewer 2 Report

Dear authors,

I have reviewed your article entitled "Gene Expression Profiling of Fibroepithelial Lesions of the Breast." Overall, I find the study to be well-designed and written, with interesting findings that could have clinical implications. However, there are several points that need to be addressed before the article can be considered for publication.

(Introduction)

1-      I would like you to include a brief background on fibroepithelial lesions of the breast and their clinical significance. Additionally, please provide a more detailed explanation of the aim of the study, including the specific research questions that you sought to address.

(Methods)

2-      I would like to see a more detailed description of the patient sample selection process, including the inclusion and exclusion criteria used in the study. Please consider including a table summarizing these criteria to aid readers in understanding the study design.

3-      Additionally, I found the methods section to be lacking in detail regarding the bioinformatic processes used in the study, such as clustering and pathway analysis. Please provide a more detailed description of these methods to aid readers in understanding your analysis.

(Results and discussion)

4-      I would also like to see figures for immunohistochemical staining. And also discuss the reasons for your selection of CD163, CD84, and CD68 while ignoring CD86. Please address this issue in the discussion section.

5-      Lastly, while the study provides interesting findings regarding gene expression profiles in FELs, it would be beneficial to confirm the genes involved in biological processes such as hypoxia-related genes, cancer-associated intracellular signaling cascades, or apoptosis using qPCR. Please consider evaluating one family of genes related to one of the inflammatory biological processes to strengthen your results.

6-      Please compare your results and methods with those presented in the study titled 'Gene expression-based classifications of fibroadenomas and phyllodes tumors of the breast' authored by Vidal et al, with more details.

Overall, your article has the potential to make a significant contribution to the field of breast cancer research, but I believe that these revisions are necessary before publication. I look forward to seeing your revised manuscript.

 Sincerely,

Author Response

1-      I would like you to include a brief background on fibroepithelial lesions of the breast and their clinical significance. Additionally, please provide a more detailed explanation of the aim of the study, including the specific research questions that you sought to address.

Additional context has been added

2-      I would like to see a more detailed description of the patient sample selection process, including the inclusion and exclusion criteria used in the study. Please consider including a table summarizing these criteria to aid readers in understanding the study design.

Selection criteria was largely based off RNA viability rather than patient characteristics; we prioritized more recent cases with better RNA preservation and that had good representative areas on a single block to optimize gene expression results.

3-      Additionally, I found the methods section to be lacking in detail regarding the bioinformatic processes used in the study, such as clustering and pathway analysis. Please provide a more detailed description of these methods to aid readers in understanding your analysis.

Additional information has been provided

4-      I would also like to see figures for immunohistochemical staining. And also discuss the reasons for your selection of CD163, CD84, and CD68 while ignoring CD86. Please address this issue in the discussion section.

Additional figure with IHC has been added.

5-      Lastly, while the study provides interesting findings regarding gene expression profiles in FELs, it would be beneficial to confirm the genes involved in biological processes such as hypoxia-related genes, cancer-associated intracellular signaling cascades, or apoptosis using qPCR. Please consider evaluating one family of genes related to one of the inflammatory biological processes to strengthen your results.

While this would be very interesting to evaluate, we are currently limited by amount of viable RNA present in our tissue blocks for additional testing.

6-      Please compare your results and methods with those presented in the study titled 'Gene expression-based classifications of fibroadenomas and phyllodes tumors of the breast' authored by Vidal et al, with more details.

Article, including results and methods, is reviewed

Round 2

Reviewer 2 Report

Dear Authors,

I have received and evaluated the revised version of your manuscript titled "Gene Expression Profiling of Fibroepithelial Lesions of the Breast". I appreciate your effort in addressing the reviewers' comments and making the necessary changes in the manuscript.

Although you were not able to conduct the experiments I have suggested, I believe the revised manuscript still contains valuable scientific information that is worth publishing in the International Journal of Molecular Sciences.

Thank you for your contribution to the scientific community.

Best regards,